# Long Non-Coding RNAs as Potential Regulators of EMT-Related Transcription Factors in Colorectal Cancer—A Systematic Review and Bioinformatics Analysis

**DOI:** 10.3390/cancers14092280

**Published:** 2022-05-03

**Authors:** Ana Pavlič, Nina Hauptman, Emanuela Boštjančič, Nina Zidar

**Affiliations:** Faculty of Medicine, Institute of Pathology, University of Ljubljana, 1000 Ljubljana, Slovenia; ana.pavlic@mf.uni-lj.si (A.P.); nina.hauptman@mf.uni-lj.si (N.H.); emanuela.bostjancic@mf.uni-lj.si (E.B.)

**Keywords:** long non-coding RNA, lncRNA, EMT, colon cancer, CRC, EMT-TF

## Abstract

**Simple Summary:**

Emerging evidence highlights long non-coding RNAs as important regulators of epithelial–mesenchymal transition. Numerous studies have attempted to define their possible diagnostic, prognostic and therapeutic values in various human cancers. The aim of this review is to summarize long non-coding RNAs involved in the regulation of epithelial–mesenchymal transition in colorectal carcinoma. Additional candidate long non-coding RNAs are identified through a bioinformatics analysis.

**Abstract:**

Epithelial–mesenchymal transition (EMT) plays a pivotal role in carcinogenesis, influencing cancer progression, metastases, stemness, immune evasion, metabolic reprogramming and therapeutic resistance. EMT in most carcinomas, including colorectal carcinoma (CRC), is only partial, and can be evidenced by identification of the underlying molecular drivers and their regulatory molecules. During EMT, cellular reprogramming is orchestrated by core EMT transcription factors (EMT-TFs), namely *ZEB1/2*, *TWIST1/2*, *SNAI1* (*SNAIL*) and *SNAI2* (*SLUG*). While microRNAs have been clearly defined as regulators of EMT, the role of long non-coding RNAs (lncRNAs) in EMT is poorly defined and controversial. Determining the role of lncRNAs in EMT remains a challenge, because they are involved in a number of cellular pathways and are operating through various mechanisms. Adding to the complexity, some lncRNAs have controversial functions across different tumor types, acting as EMT promotors in some tumors and as EMT suppressors in others. The aim of this review is to summarize the role of lncRNAs involved in the regulation of EMT-TFs in human CRC. Additional candidate lncRNAs were identified through a bioinformatics analysis.

## 1. Introduction

Epithelial–mesenchymal transition (EMT) is a process of cellular transdifferentiation whereby stationary epithelial cells acquire a motile mesenchymal phenotype [1]. It is generally accepted that EMT is an important mechanism in human carcinogenesis, but despite extensive research, its role in cancer development and progression is not completely understood yet. Cells undergoing EMT gain the ability to migrate and invade beyond the invasive tumor front, promoting tumor progression and metastasis [2]. Additionally, studies indicate that EMT plays a pivotal role in cancer cell stemness, immune evasion, metabolic reprogramming and therapeutic resistance [3,4].

Full EMT involves a complete transition from an epithelial to a mesenchymal cell, evidenced by an altered morphology and a change in EMT-related molecular biomarkers [5,6]. Full EMT is rare in human cancers, but it does occur, for example in spindle cell carcinoma and carcinosarcoma [7]. EMT in most other carcinomas including colorectal cancer (CRC) is only partial, giving rise to a spectrum of intermediate cellular transition states [8,9]. Traditional biomarkers based on protein expression characteristic of either epithelial (E-cadherin, cytokeratins) or mesenchymal (N-cadherin, vimentin, fibronectin) cells are insufficient for defining the various states of partial EMT. Therefore, evidence of partial EMT relies on identification of molecular drivers governing EMT, transcription factors and their regulatory molecules [10,11].

EMT is orchestrated by EMT transcription factors (EMT-TF), which can both repress epithelial genes involved in cellular adhesion, polarity and cytoskeleton reorganization as well as activate genes associated with a mesenchymal phenotype. Among the most investigated EMT-TFs are *ZEB1/2*, *TWIST1/2*, *SNAI1* (*SNAIL*) and *SNAI2* (*SLUG*). These are crucial for maintaining normal epithelium structure and their dysregulation induces EMT [12,13,14,15]. Over the past years, studies on various different cancers in vitro and in vivo also highlighted their role in other aspects of tumorigenesis, such as regulation of stemness, therapy resistance, immune evasion, DNA integrity and metabolic reprogramming. EMT-TFs usually cooperate with one another and likely act in a tumor- and dosage-specific manner, consequently exhibiting different expression patterns and functions across tumor types [3,16,17,18]. Their expression is regulated on multiple cellular levels including non-coding RNAs as one of the epigenetic mechanisms. While microRNAs have been clearly defined as regulators of EMT, less is known about the role of long non-coding RNAs (lncRNAs) in EMT in human cancers [19,20,21,22].

LncRNAs have emerged as important regulators of EMT in human cancers and an increasing number of studies have attempted to define their possible diagnostic, prognostic and therapeutic values. LncRNAs are RNA transcripts longer than 200 base pairs that do not encode proteins. Instead, they influence various cellular processes by different modes of action: as mediators of gene transcription; as decoy molecules for proteins, which results in dysregulation of DNA and mRNA; as partners in competing endogenous RNA (ceRNA), which results in miRNA depletion and increasing expression of their targeted genes; as guide molecules for proteins; and as scaffold molecules resulting in the assembly of macromolecular complexes that regulate the expression of their target genes [23,24].

LncRNAs may play a critical role in tumor progression and metastasis due to their ability to regulate EMT-TFs. The regulation can take place in the nucleus at epigenetic or transcription level or as regulation of pre-mRNA alternative splicing. The regulation can also take place in the cytoplasm at the post-transcriptional level, which includes mRNA translation and stability, protein stability and the ceRNA network [24]. However, because their regulation is diverse, determining the effect of lncRNAs on the process of EMT remains a challenge [21,24,25]. Adding to the complexity, some lncRNAs have controversial functions across different tumor types, acting as EMT promotors in some tumors and as EMT suppressors in others [26,27,28,29].

A thorough understanding of EMT-TF regulation is a prerequisite to defining complete and partial EMT. For this reason, we aim to investigate the role of lncRNAs involved in the regulation of EMT-TFs in human CRC. The study provides candidate lncRNAs identified through a systematic literature review as well as by performing a bioinformatics analysis.

## 2. Methods

### 2.1. Systematic Literature Review

#### 2.1.1. Study Design

This study was conducted using the “Preferred Reporting Items for Systematic Reviews and Meta-analyses” (PRISMA) guidelines [30]. The study is registered with the Research Registry and its unique identifying number (UIN) is: reviewregistry1311. The online PubMed (NCBI, Bethesda, MD, USA) database was systematically searched for all articles relating to lncRNAs and their role in relation to EMT-TFs in colorectal adenocarcinoma for the period from 1 January 1990 to 30 March 2022. No language restrictions were applied. The search was conducted using the following terms: »colorectal«, »colon«, »long non-coding« and »lncRNA« in combination with each of the following transcription factors: »SNAIL«, »SNAI1«, »SLUG«, »SNAI2«, »TWIST«, »TWIST1«, »TWIST2«, »ZEB«, »ZEB1« and »ZEB2« present in either the title or the abstract of articles. »AND« was used to connect main research terms whereas »OR« was used to incorporate synonym words.

#### 2.1.2. Inclusion and Exclusion Criteria

Studies were included if they provided insight into the molecular signaling between lncRNAs and any of the core EMT-TFs: *SNAI1* (*SNAIL*), *SNAI2* (*SLUG*), *TWIST1* (*TWIST*), *TWIST2*, *ZEB1* or *ZEB2* in human CRC samples, in vitro human CRC cell lines and/or animal models. Studies were excluded if (i) they did not pertain to CRC, (ii) they were focused on other signaling pathways without evaluating the expression of the above-mentioned TFs and/or lncRNAs (iii) they were reviews, conference abstracts, unpublished manuscripts and letters, or (iv) full-text articles were not available.

#### 2.1.3. Data Extraction and Outcomes

Following the conducted search, the titles and abstracts of articles were screened by two reviewers (AP and NZ). For the sake of completion, the references of the articles were manually searched for additional studies that, based on their titles, appeared relevant to this review, but were not found with the primary database search. These articles were added to the screening process and were then further categorized based on the inclusion and exclusion criteria with the rest of the articles. The following data were extracted from each article: authors, year and journal of publication, patient characteristics (overall survival, TNM stage, lymphovascular invasion), model type (human tissue, CRC cell lines, experimental animals), main study findings including expression of lncRNA and EMT-TF in CRC, function and the proposed mechanism of lncRNA in CRC.

### 2.2. Bioinformatics Analysis

A bioinformatics analysis of lncRNA and TF correlations on TCGA datasets of colon adenocarcinoma and rectum adenocarcinoma was performed using The Cancer Genome Atlas (TCGA) with TCGAbiolinks package in R programming language [31].

## 3. Results

Eighty-five articles were identified through a PubMed database search. Duplicates were removed (*n* = 16). Additional articles pertaining to this topic (*n* = 11) were manually searched from the references of records identified through the database search. They were screened along with those identified through the database search to assess their relevance, as some contained relevant data despite not being found with the initial search. In total, eighty articles were screened. Thirty-three were excluded due to the following reasons: research on cancers other than colorectal (*n* = 10), ineligible because the TFs and/or lncRNAs of interest were not evaluated (*n* = 16), retracted article (*n* = 1), review article (*n* = 3), unavailable article (*n* = 1), analysis of treatment effect (*n* = 1) and bioinformatics analysis (*n* = 1), leaving 47 studies for inclusion in the final analysis. A flow chart of the article identification and selection process is provided in Figure 1.

### 3.1. LncRNAs Regulating a Single EMT-TF

Fourty-one lncRNAs acting as regulators of one or more EMT-TFs were identified. Twenty-five lncRNAs were shown to regulate a single EMT-TF; *ZEB1* (*n* = 13), *SNAI1* (*n* = 4), *SNAI2* (*n* = 3) and *TWIST1* (*n* = 5). A short summary of their cellular effects is provided in Table 1. None of the lncRNAs were shown to influence the expression of ZEB2 only.

### 3.2. ZEB1

The greatest number of studies researching lncRNAs as regulators of EMT-TFs pertained to *ZEB1*. Twenty lncRNAs modulating the activity of *ZEB1* have been identified so far. All exhibit a pro-EMT effect. Thirteen lncRNAs were shown to influence the expression of *ZEB1* only. Their activation was mostly associated with a shorter overall survival and a higher TNM stage. In vitro studies were performed for all 13 lncRNAs. Apart from *DUXAP8*, which was shown to promote cell proliferation only, the effects of all other lncRNAs were similar as observed in vitro. They mainly affected cell migration and, in many cases, also cell invasion. Metastatic potential was additionally confirmed in vivo for five out of seven lncRNAs (*AC010789.1*, *AK000053*, *LINC01413*, *N-BLR*, *RP11* and *XIST*). Even though *ZFAS* exhibited migratory and invasive traits in vitro, it only promoted tumor growth and not metastasis formation in vivo. Interestingly, *AK000053* and *RP11* were also found to influence cell morphology promoting a mesenchymal phenotype, further highlighting their role in EMT. *AK000053* was the only lncRNA from this group, which was also shown to evoke stem cell properties.

### 3.3. SNAI1

*SNAI1* is frequently regulated alongside other TFs as a result of various lncRNA activations (*n* = 13), whereas only four lncRNAs seem to regulate *SNAI1* only. Two of them, namely *GNAT1-1* and *SATB2-AS1*, exhibited an anti-EMT effect. Their activation was associated with an increased overall patient survival and a lower TNM stage. Both were also shown to decrease migration and cell invasion in vitro and decrease the incidence of metastasis in vivo. The other two lncRNAs modulating *SNAI1*, namely *LDLRAD4-AS1* and *TRERNA1*, exhibited a pro-EMT effect, with cellular migration and invasion observed in vitro. They were both associated with nodal and distant metastases in CRC patients. High expression of *TRERNA1* was also associated with metastatic traits in vitro; however, metastatic potential in vivo was confirmed only for *LDLRAD4-AS1*.

### 3.4. TWIST1

Five lncRNAs (*AK027294*, *CHRF*, *LINCOO467*, *LINC00941* and *SNHG11*) were shown to stimulate EMT by influencing the expression of *TWIST1* only. All of them were associated with an increased migratory and/or invasive ability of cells in vitro. In vivo studies were available for *CHRF*, *LINC00941* and *SNHG11*, all of which were shown to be associated with an increased metastatic potential. In addition, high expression of *CHRF* and *LINC00941* also triggered a change in cell morphology, a feature of a complete EMT.

### 3.5. SNAI2

*SNAI2* was solely regulated by three lncRNAs, namely *GAPLINC*, *SNHG15* and *XLOC_010588*, all exerting a pro-EMT effect, with cellular migration and/or invasion observed in vitro. Interestingly, none of these lncRNAs promoted metastasis formation either in vitro or in vivo.

### 3.6. LncRNAs Regulating Multiple EMT-TFs

Sixteen lncRNAs were shown to regulate either two (*n* = 9) or three (*n* = 7) EMT-TFs (Table 2). The identified lncRNAs and a short summary of their effects are provided in Table 3.

### 3.7. ZEB1 and ZEB2

The effects of lncRNAs regulating both *ZEB1* and *ZEB2* (*DLEU2*, *LINC01296*) were similar to those of lncRNAs regulating *ZEB1* alone. Both correlated with distant metastasis in patients with CRC and a cellular invasive phenotype observed in vitro. As of yet, no functional analyses have been performed in vivo.

### 3.8. ZEB1 and SNAI1

Two lncRNAs were shown to regulate both *ZEB1* and *SNAI1*, namely *VIM-AS1* and *ZEB2-AS1*. While they were both associated with a shorter survival, observed in vitro effects were different. *VIM-AS1* promoted cellular proliferation and migration and had an effect on cell morphology, while *ZEB2-AS1* was linked to cell migration and invasion only. Their effects have not yet been investigated in vivo.

### 3.9. ZEB1, ZEB2 and SNAI1

*H19* has been shown to affect the expression of *ZEB1*, *ZEB2* and *SNAI1*. Expression of *H19* in CRC correlated with a shorter survival and an increased TNM stage. The effects of *H19* in vitro were assessed by two study groups, both reporting an increased cellular proliferation and migration of CRC cells. One of the groups also detected a change in cell morphology while the other observed an invasive and metastasizing cellular phenotype. *H19* has been shown to stimulate tumor growth in vivo.

### 3.10. ZEB1, SNAI1 and SNAI2

*SNHG6* and *XIST* have been found to influence the expressions of *ZEB1*, *SNAI1* and *SNAI2* in CRC. Both have been linked to a decreased overall survival. Their expression promoted proliferation, migration and invasion of cancer cells in vitro. *SNHG6* also evoked metastatic cellular traits in vitro; however, its metastatic potential was not confirmed in vivo. On the contrary, higher expression of *XIST* both increased metastasis formation in vivo and evoked stem cell properties and the spindle morphology of cells in vitro.

### 3.11. ZEB2, SNAI1 and SNAI2

*UICLM* influenced the expression of *ZEB2*, *SNAI1* and *SNAI2*. Its expression was associated with a shorter overall survival and a higher TNM stage. In vitro, it stimulated cell proliferation and invasion as well as promoted stem cell properties. In vivo, its expression increased tumor size and promoted metastasis formation.

### 3.12. ZEB2 and SNAI2

*B3GALT5-AS1* regulating *ZEB2* and *SNAI2* has been shown to have an anti-EMT effect in CRC. Its expression was associated with an increased overall patient survival and inhibited migratory and invasive cell abilities in vitro. In vivo studies also suggest that high *B3GALT5-AS1* expression prevents metastasis formation.

### 3.13. ZEB2, SNAI1 and TWIST1

The effect of *TUG1* on EMT-TF in CRC has been investigated by two groups. One found that *TUG1* augmented proliferation and the invasive traits of CRC cells in vitro by regulating *ZEB2* and *SNAI1*. The other observed no increase in cancer cell proliferation in vitro, but rather an increase in migratory as well as invasive cell abilities. By regulating *TWIST1* they also showed that *TUG1* expression influences metastasis formation in vivo.

### 3.14. SNAI1 and SNAI2

*MAPKAPK5-AS1* and *HOTAIR* have been shown to promote EMT in CRC by regulating *SNAI1* and *SNAI2*. Their expression exerted a similar effect. Both lncRNAs promoted cell migration and invasion in vitro and stimulated the formation of distant metastasis in vivo. The expression of *MAPKAPK5-AS1* has also been linked to an increased TNM stage and a shorter overall survival, whereas this research has not been conducted for *HOTAIR*.

### 3.15. SNAI1 and TWIST1

*SNAI1* and *TWIST1* are regulated by lncRNAs *CTD-903* and *PANDAR*. *CTD-903* was shown to suppress EMT in vitro, as evidenced by decreased cell proliferation, migration and invasion. It also promoted an epithelioid cellular phenotype. The expression of *PANDAR* increased cell proliferation, migration and invasion in vitro. The in vivo effects of neither *CTD-903* nor *PANDAR* have been investigated.

### 3.16. SNAI1, SNAI2 and TWIST1

*HOXA11-AS* and *MIR4435-2HG* have been shown to regulate three EMT-TFs, namely *SNAI1*, *SNAI2* and *TWIST1*. High expression of both *HOXA11-AS* and *MIR4435-2HG* was associated with a higher rate of metastasis in CRC patients and they were both shown to evoke migration and invasion of cells in vitro. Additionally, *MIR4435-2HG* also promoted tumor growth and distant metastasis formation in vivo, whereas such studies have not been performed yet for *HOXA11-AS*.

### 3.17. Bioinformatics Analysis

Data were available for 609 tumor samples and 51 normal samples. We obtained a list of 18,855 known lncRNAs from (http://genome.igib.res.in/lncRNome, accessed on 27 January 2022). The intersect between the list of known lncRNAs and TCGA data yielded 3418 lncRNAs. For the purpose of this study, we calculated Pearson’s correlation with the package “sigr” among all the lncRNAs included in TCGA and EMT-TFs: *ZEB1*, *ZEB1*, *SNAI1*, *SNAI2*, *TWIST1* and *TWIST2* [78].

Significant correlations (*p* < 0.05) were divided into groups designated as “very strong”, “strong”, “moderate” and “weak” correlations, based on correlation coefficients of at least |0.8|, |0.6|, |0.4|, and |0.2|, respectively. The analysis yielded 258 significant correlations of lncRNAs-TF pairs in tumor samples (*n* = 2 very strong; *n* = 11 strong; *n* = 41 moderate; *n* = 204 weak). The significant strong and very strong correlations are presented in Table 4, while all the results are presented in Appendix A.

## 4. Discussion

The aim of this review was to summarize and identify lncRNAs involved in the process of EMT in CRC and examine their functions. Through a systematic literature review and a bioinformatics analysis, we identified lncRNAs shown to regulate core EMT-TFs responsible for cellular reprogramming during EMT in CRC, namely *ZEB1/2*, *TWIST1/2* and *SNAI1/2*.

Through a systematic literature review, we identified 41 lncRNAs shown to regulate one or more core EMT-TFs in CRC. With regard to clinical relevance, the expression of the most lncRNAs significantly correlated with overall survival and TNM stages. The majority of lncRNAs were upregulated in CRC and promoted the “classical” traits of EMT. Following their knock-out in vitro and in vivo, the proliferative, migratory and invasive capabilities of cancer cells diminished. Frequently, the expression of investigated lncRNAs also enhanced the metastatic potential of CRC cells. Importantly, the effects of lncRNAs in CRC cell lines did not always correlate to the ones on animal models. Studies containing both in vitro and in vivo results were available for 24 lncRNAs. Most of lncRNAs were shown to trigger invasive cancer cell traits in vitro and concordantly promoted metastasis formation in vivo. A few lncRNAs were not associated with cell invasion in vitro or increased metastatic potential in vivo. By contrast, certain lncRNAs (*ZFAS1*, *GAPLINC*, *SNHG6*) promoted invasive or even metastatic potential on CRC cells in vitro and yet were not observed to enhance the metastatic potential of tumors in vivo [45,50,71]. Therefore, conclusions regarding the role of lncRNAs in CRC progression should not be based solely on in vitro studies as further validation is required before establishing any possible prognostic role of lncRNAs in the clinical setting.

Other features of EMT were less frequently associated with lncRNAs. Firstly, silencing certain lncRNAs prompted a change in cell morphology, a feature of full EMT [26,34,42,54,56,59,61,76]. This is consistent with previous studies linking *ZEB1*, *SNAI1* and *TWIST1* with a mesenchymal phenotype of tumor cells in CRC. *ZEB1* was overexpressed particularly in tumor cells at the invasive front and its knock-down promoted an epithelioid phenotype in CRC [79]. In CRC cells overexpressing *SNAI1*, a clear switch to a mesenchymal phenotype was observed microscopically [80]. The presence of *TWIST1* expression in CRC was reported mainly in cancer cells with a mesenchymal phenotype located in the tumor stroma. Additionally, cell lines transfected with *TWIST1* acquired characteristics of activated cancer-associated fibroblasts [81,82]. Secondly, *AK000053*, *XIST* and *UICLM* were shown to evoke stem cell properties in CRC [26,34,75]. This corroborates previous findings linking their EMT-TF targets (*ZEB1*, *SNAI1* and *SNAI2*) with cancer stem cells in CRC. *SNAI1* was shown to induce a stem-cell-like phenotype [80,83], while *ZEB1* [84] and *SNAI2* [85] were suggested to play critical roles in the maintenance of self-renewal capacity in CRC. Lastly, some studies also examined the role of lncRNAs in therapy resistance. By affecting the expression of *ZEB1*, *N-BLR* and *LINC00460* induced chemoresistance and contributed to the radioresistance of CRC, respectively [39,41]. This is consistent with previous findings indicating that *ZEB1* plays a critical role in the EMT-regulated oxaliplatin resistance of CRC [86].

It is important to note that none of the investigated lncRNAs were shown to influence *ZEB2* only. We can only speculatively suggest possible explanations. Firstly, similarly to squamous cell carcinoma of the head and neck, *ZEB1* and *ZEB2* could be co-expressed and regulated together [87]. Secondly, it is possible that *ZEB2* activation occurs incidentally during the regulation of *ZEB1* or *SLUG*, as frequently observed with miRNAs [88]. Thirdly, TFs might activate and co-regulate each other to provide a certain phenotype, similarly to stem cell markers or members of one miRNA family. The activation of the EMT program might depend on the co-regulation of different EMT-TFs (e.g., *SNAI2* and *ZEB2*) [89]. Lastly, lncRNAs influencing only *ZEB2* have yet to be discovered.

Accurate evaluation of the role of lncRNAs in EMT requires further research taking into account their possible differential expression in the tumor microenvironment. The majority of studies identified in this review utilized whole-tumor RNA isolation, thus disregarding the important concept of intra-tumor heterogeneity. While the genetic landscape of the entire tumor may be very similar, the distribution and frequency of epigenetic changes vary within a tumor [90]. Accordingly, it has been shown that partial EMT is mainly activated at the invasive tumor front, especially in the areas of basement membrane disintegration and in the areas of tumor budding [91,92,93,94]. Assessing the expression of lncRNAs at these critical sites could provide insight into the mechanism of their formation and possibly clarify their role in cancer progression.

The bioinformatics analysis uncovered 258 significant correlations of lncRNAs to EMT-TFs on colon and rectum adenocarcinoma datasets. Among the lncRNAs with either strong or very strong correlations to EMT-TFs, only two lncRNAs, namely *MAGI2-AS3* and *TP73-AS1* have been further validated on CRC cell lines and mouse models [95,96,97]. *MAGI2-AS3* promoted CRC progression by facilitating the proliferation and migration of cancer cells in vitro and promoted tumor growth in vivo [95]. Similarly, *TP73-AS1* promoted the proliferation, migration and invasion of cancer cells in vitro as well as tumor growth in vivo [96,97]. Overexpression of *TP73-AS1* was also associated with metastasis and advanced clinical stages in CRC patients [97]. Interestingly, only *ZEB1*, *ZEB2* and *SLUG* had strong or very strong correlations with lncRNAs. We can only speculate the reason for this finding. Certain EMT-TFs may have different functions in different types of tissue or may be regulated by other factors with varying intensity. Therefore, it is possible that *ZEB1*, *ZEB2* and *SLUG* are involved in the regulation of EMT in CRC more strongly than other EMT-TFs, or the correlated lncRNAs have tissue-specific expressions influencing these three EMT-TFs more strongly than others [89,98].

The bioinformatics analysis yielded a high number of correlations between lncRNAs and EMT-TFs in CRC; however, their significance in EMT is uncertain. Any correlation, either positive or negative, suggests that EMT-TFs might be regulated by lncRNAs or vice versa. EMT-TFs are regulated either directly (binding of lncRNAs to the EMT-TF or binding of EMT-TF to the promotor of lncRNAs) or indirectly (e.g., through sponging miRNAs by lncRNAs). There are a huge number of possibilities to be explored in the regulation of EMT-TFs by lncRNAs or vice versa thus influencing the activation of EMT or leading to its partial or full EMT phenotype. In turn, all these possible regulations might be also associated with clinicopathological characteristics that are consequences of EMT [98]. Further validation in vitro and in vivo is required to evaluate their function in EMT and pinpoint those with a potential diagnostic, prognostic and therapeutic value.

There are a few limitations of this review. Firstly, all the studies providing any evidence of lncRNAs mediating an effect on one or more EMT-TFs were included, even though the exact mechanism of action of many lncRNAs was not shown, particularly in cases reporting a change in the expression of multiple EMT-TFs. It is uncertain whether they are all in fact direct targets of the investigated lncRNA or if the regulation is indirect, since it is well known that many EMT-TFs regulate each other as well [10]. Secondly, this review also focuses only on the regulation of core EMT-TFs. Other TFs have also been implicated in the regulation of EMT; however, their functions are less well established. They have either not been shown to induce EMT in cell cultures and animal models or there is insufficient information on whether they induce EMT directly or through the activation of core EMT-TFs [10,99]. Therefore, they were not included in this review.

## 5. Conclusions

Emerging evidence highlights lncRNAs as regulators of EMT, potentially making them valuable diagnostic markers or therapeutic targets and thus warranting further research. It remains to be seen if their expression patterns and mechanisms of action may contribute to the unresolved issues of EMT, particularly regarding partial EMT states.

## Figures and Tables

**Figure 1 cancers-14-02280-f001:**
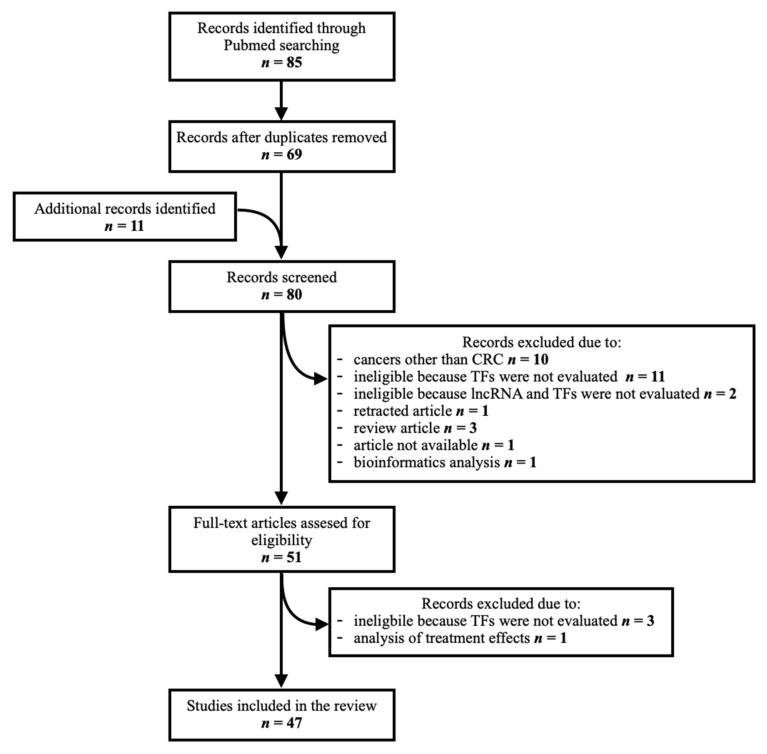
Schematic representation of article identification and selection process.

**Table 1 cancers-14-02280-t001:** Identified lncRNAs regulating a single EMT-TF and their major effects. Legend: CSC, cancer stem cell properties; Invas., invasion; LVI, lymphovascular invasion; Migr., migration; M, metastasis stage; Met., distant metastasis; Morph., change in morphology; N, node stage; OS, overall patient survival; Prolif., proliferation; T, tumor stage; ↓, decreased; ↑, increased.

lncRNA	TF	Patient Characteristics					In Vitro						In Vivo		Ref.
		OS	T	N	M	LVI	Prolif.	Migr.	Invas.	Met.	CSC	Morph.	Tumor Size	Met.	
*AC010789.1*	*ZEB1*	↓		↑			↑	↑	↑				↑	↑	[32]
*AFAP1-AS1*	*ZEB1*	↓						↑	↑				/	/	[33]
*AK000053*	*ZEB1*	↓						↑	↑		↑	yes	↑	↑	[34]
*ATB*	*ZEB1*	↓		↑		↑	↑	↑					/	/	[35]
*DUXAP8*	*ZEB1*		↑			↑	↑						↑		[36]
*HCP5*	*ZEB1*	↓	↑	↑	↑			↑	↑				/	/	[37]
*KCNQ1OT1*	*ZEB1*	↓					↑	↑					/	/	[38]
*LINC00460*	*ZEB1*	/	/	/	/	/	↑	↑					/	/	[39]
*LINC01413*	*ZEB1*	↓	↑	↑	↑		↑	↑	↑				↑	↑	[40]
*N-BLR*	*ZEB1*	↓	↑	↑	↑			↑	↑					↑	[41]
*RP11*	*ZEB1*		↑	↑	↑		no	↑	↑			yes	no	↑	[42]
*ZEB1-AS1*	*ZEB1*	↓	↑	↑	↑		↑	↑	↑				/	/	[43,44]
*ZFAS1*	*ZEB1*	↓					↑	↑	↑				↑		[45]
*GNAT1*	*SNAI1*	↑	↓	↓	↓	↓	↓	↓	↓					↓	[46]
*LDLRAD4-AS1*	*SNAI1*	↓	↑	↑	↑	↑		↑	↑					↑	[47]
*SATB2-AS1*	*SNAI1*	↑	↓	↓	↓		↓	↓	↓				↓	↓	[48]
*TRERNA1*	*SNAI1*	↓	↑	↑	↑	no		↑	↑	↑			/	/	[49]
*GAPLINC*	*SNAI2*	↓	↑	↑			↑		↑				↑		[50]
*SNHG15*	*SNAI2*	↓					no	↑					↑		[51]
*XLOC_010588*	*SNAI2*	↓	↑	↑	↑		↑	↑	↑				/	/	[52]
*AK027294*	*TWIST1*	/	/	/	/	/	↑	↑					/	/	[53]
*CHRF*	*TWIST1*	↓	↑	↑				↑	↑			yes	no	↑	[54]
*LINC00467*	*TWIST1*	↓			↑		↑		↑				/	/	[55]
*LINC00941*	*TWIST1*	↓		↑				↑	↑			yes		↑	[56]
*SNHG11*	*TWIST1*	/	/	/	/	/		↑	↑					↑	[57]

**Table 2 cancers-14-02280-t002:** LncRNAs regulating multiple EMT-TFs.

LncRNA	*ZEB1*	*ZEB2*	*SNAI1*	*SNAI2*	*TWIST1*
*DLEU2*	x	x			
*LINC01296*	x	x			
*VIM-AS1*	x		x		
*ZEB2-AS1*	x		x		
*H19*	x	x	x		
*SNHG6*	x		x	x	
*XIST*	x		x	x	
*B3GALT5-AS1*		x		x	
*TUG1*		x	x		x
*UICLM*		x	x	x	
*MAPKAPK5-AS1*			x	x	
*HOTAIR*			x	x	
*CTD-903*			x		x
*PANDAR*			x		x
*HOXA11-AS*			x	x	x
*MIR4435-2HG*			x	x	x

**Table 3 cancers-14-02280-t003:** Identified lncRNAs regulating multiple EMT-TFs and their major effects. Legend: CSC, cancer stem cell properties; Invas., invasion; LVI, lymphovascular invasion; Migr., migration; M, metastasis stage; Met., distant metastasis; Morph., change in morphology; N, node stage; OS, overall patient survival; Prolif., proliferation; T, tumor stage; ↓, decreased; ↑, increased.

LncRNA	TFs	Patient Characteristics					In Vitro						In Vivo		Ref.
		OS	T	N	M	LVI	Prolif.	Migr.	Invas.	Met.	CSC	Morph.	Tumor Size	Met.	
*B3GALT5-AS1*	*ZEB2 SNAI2*	↑			↓		↓	↓	↓					↓	[58]
*CTD-903*	*SNAI1 TWIST1*	↑	no	no	no		↓	↓	↓			yes	/	/	[59]
*DLEU2*	*ZEB1 ZEB2*	↓			↑		↑		↑				/	/	[60]
*H19*	*ZEB1 ZEB2*	/	/	/	/	/	↑	↑				yes	↑		[61]
*H19*	*ZEB1*	↓	↑	↑	↑	↑	/	/	/	/	/	/	/	/	[62]
*H19*	*SNAI1*	↓	↑	↑	↑	↑	↑	↑	↑	↑			/	/	[63,64]
*HOTAIR*	*SNAI1 SNAI2*	/	/	/	/	/		↑	↑				↑	↑	[65]
*HOXA11-AS*	*SNAI1 SNAI2 TWIST1*	↓			↑			↑	↑				/	/	[66]
*LINC01296*	*ZEB1 ZEB2*	↓	↑	↑	↑			↑	↑				/	/	[67]
*MAPKAPK5-AS1*	*SNAI1 SNAI2*	↓	↑	↑	↑		↑	↑	↑					↑	[68]
*MIR4435-2HG*	*SNAI1 SNAI2 TWIST1*	↓	↑	↑	↑	no	↑	↑	↑	↑			↑	↑	[69]
*PANDAR*	*SNAI1 TWIST1*	↓	↑	↑	no		↑	↑	↑				/	/	[70]
*SNHG6*	*ZEB1 SNAI1 SNAI2*	↓					↑	↑	↑	↑			↑		[71]
*SNHG6*	*SNAI1*	/	/	/	/	/		↑	↑				/	/	[72]
*TUG1*	*ZEB2 SNAI1*	/	/	/	/	/	↑		↑				/	/	[73]
*TUG1*	*TWIST1*	/	/	/	/	/	no	↑	↑					↑	[74]
*UICLM*	*ZEB2 SNAI1 SNAI2*	↓	↑	↑	↑		↑		↑		↑		↑	↑	[75]
*VIM-AS1*	*ZEB1 SNAI1*	↓	no	↑		↑	↑	↑				yes	/	/	[76]
*ZEB2-AS1*	*ZEB1 SNAI1*	↓						↑	↑				/	/	[77]
*XIST*	*ZEB1 SNAI1 SNAI2*	↓	↑	↑	↑		↑	↑	↑		↑	yes	↑	↑	[26]

**Table 4 cancers-14-02280-t004:** Statistically significant correlations of selected lncRNAs and EMT-TFs in tumor of TCGA colon and rectum adenocarcinoma datasets. ↑↑↑, very strong positive correlation; ↑↑, strong positive correlation.

LncRNA	*ZEB1*	*ZEB2*	*SNAI2*
*AC106786.1*	↑↑	↑↑	
*AC112721.2*		↑↑	↑↑
*ADAMTS9-AS1*	↑↑		
*ADAMTS9-AS2*	↑↑↑		
*DNM3OS*	↑↑	↑↑	↑↑
*FNDC1-IT1*		↑↑	
*LINC00578*	↑↑	↑↑	
*MAGI2-AS3*	↑↑↑	↑↑↑	↑↑
*PIK3R6*		↑↑	
*TP73-AS1*	↑↑	↑↑	
*VCAN-AS1*		↑↑	
*ZNF154*	↑↑	↑↑

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
