# Peer review of "Long Non-Coding RNAs as Potential Regulators of EMT-Related Transcription Factors in Colorectal Cancer—A Systematic Review and Bioinformatics Analysis"

_cancers, 2022, doi:10.3390/cancers14092280_

Round 1
Reviewer 1 Report
This topic has various similar publications inside the literature.
Author Response
Comment #1: This topic has various similar publications inside the literature.
- No doubt this is an emerging and very relevant topic with a number of publications attempting to define the role of lncRNAs in various tumor types. Given the complexity and sometimes even opposite functions in different cancers, we aimed to summarize what is currently known about the regulation of lncRNAs in EMT in CRC. In addition, we present new lncRNAs, potentially relevant in EMT in CRC, found through bioinformatics analysis.
Reviewer 2 Report
In the present manuscript, Pavlic et. al. has reviewed recent important findings regarding long non-coding RNAs involved in the regulation of epithelial-mesenchymal transition in colorectal carcinoma. Moreover, using bioinformatics tools, they have introduced a number of candidate long non-coding RNAs which might contribute to the pathology of CC.
The review has been written very well and covers most aspects of Lnc's findings related to CC.
There are a few tipos which need to look after.
The references need to be updated with 2022 published articles and findings.
Author Response
Comment #1: There are a few tipos which need to look after.
- The manuscript was now corrected by a native English speaker.
Comment #2: The references need to be updated with 2022 published articles and findings.
- Thank you for this comment. We made an updated search on March 30th 2022 to found possible new articles and added 3 additional articles that were overlooked during the first search. We apologize for this mistake.
Reviewer 3 Report
The manuscript “Long non-coding RNAs as potential regulators of EMT-related transcription factors in colorectal cancer – a systematic review and bioinformatics analysis” by Pavlič et al. aims to summarize the current evidence for regulation of EMT-inducing transcription factors (EMT-TFs) by long non-coding RNAs (lncRNAs) in colorectal cancer (CRC).
As such, this is a highly interesting topic. However, several key issues should be addressed before being acceptable for publication.
The introduction does not provide any background about lncRNAs. This has to be changed.
In its current state, the manuscript does not provide any specific evidence for EMT-TF regulation by lncRNAs. While the authors emphasize their bibliographical literature-mining approach which aims to give a general overview over the publications addressing the lncRNA-EMT-TF connection in CRC, what is missing from the manuscript is a more detailed description of the molecular mechanisms how the mentioned lncRNAs in fact could regulate EMT-TFs, backed up by specific examples. Even though the manuscript title takes into account this limitation (“potential”), more specific examples for lncRNA-EMT-TF regulation would greatly improve the manuscript. A figure would be nice in that regard.
Furthermore, the authors show a bioinformatics analysis of RNA-Seq data from TCGA which aims to corroborate the potential regulation of EMT-TFs by lncRNAs. However, a positive expression correlation between individual lncRNAs and EMT-TFs, while interesting, does not necessarily indicate a direct regulatory relationship. Moreover, it could indicate (transcriptional) regulation of lncRNAs by EMT-TFs. This possibility should at least be mentioned. The same applies to the association of lncRNAs with certain clinicopathological characteristics that could be related to an EMT phenotype, and the effects of lncRNAs on cell migration and invasion in vitro.
Minor points :
Line 31 : “It is generally accepted that EMT plays a key role in human carcinomas, but its role remains controversial.” This sentence has to be re-phrased and elaborated on. In its current state, this statement is contradictory in itself.
Line 54 : “A thorough understanding of EMT-TF regulation is a prerequisite in defining partial EMT.” Why specifically partial EMT ? This has to be elaborated on.
Author Response
Comment #1: The introduction does not provide any background about lncRNAs. This has to be changed.
- We added the description of the overall function and relevance of lncRNA in the Introduction as you suggested.
Comment #2: In its current state, the manuscript does not provide any specific evidence for EMT-TF regulation by lncRNAs. While the authors emphasize their bibliographical literature-mining approach which aims to give a general overview over the publications addressing the lncRNA-EMT-TF connection in CRC, what is missing from the manuscript is a more detailed description of the molecular mechanisms how the mentioned lncRNAs in fact could regulate EMT-TFs, backed up by specific examples. Even though the manuscript title takes into account this limitation (“potential”), more specific examples for lncRNA-EMT-TF regulation would greatly improve the manuscript. A figure would be nice in that regard.
- Regarding specific examples of regulation, we believe the topic is very broad and difficult to summarize and it would greatly prolong the manuscript. Moreover, one of our co-authors (N. Hauptman) recently published a manuscript summarizing the regulatory role of lncRNAs sponging miR-200 family. Therefore, we would like to avoid duplication of publication. Moreover, our aim was to summarize all the possible lncRNAs that could influence the expression of the canonical EMT-TFs thus showing the huge amount of data that have yet to be explored. We believe that even if the specific mechanisms are not included, our manuscript provides information that may be helpful for future research in the field.
- Following you suggestion, we prepared a figure submitted now as a graphical abstract. Alternatively, it could also be included in the manuscript.
Comment #3: The authors show a bioinformatics analysis of RNA-Seq data from TCGA which aims to corroborate the potential regulation of EMT-TFs by lncRNAs. However, a positive expression correlation between individual lncRNAs and EMT-TFs, while interesting, does not necessarily indicate a direct regulatory relationship. Moreover, it could indicate (transcriptional) regulation of lncRNAs by EMT-TFs. This possibility should at least be mentioned. The same applies to the association of lncRNAs with certain clinicopathological characteristics that could be related to an EMT phenotype, and the effects of lncRNAs on cell migration and invasion in vitro.
- We agree with this comment. Moreover, any correlation, either positive or negative suggests that EMT-TF might be regulated by lncRNAs or vice versa. They are regulated either directly (binding of lncRNAs to the EMT-TF or binding of EMT-TF to the promotor of lncRNAs) or indirectly (g., through sponging miRNAs by lncRNAs). However, our aim was to show that there is a huge number of possibilities to be explored in regulation of EMT-TFs by lncRNAs or vice versa and as such influencing activation of EMT leading to partial or full EMT program. In turn, all this possible regulations might be also associated with clinicopathological characteristics that are consequences of EMT. We added this in the manuscript.
Comment #4: Line 31 : “It is generally accepted that EMT plays a key role in human carcinomas, but its role remains controversial.” This sentence has to be re-phrased and elaborated on. In its current state, this statement is contradictory in itself.
- We changed this sentence to »It is generally accepted that EMT is an important mechanism in human carcinogenesis, but despite extensive research, its role in cancer development and progression is not completely understood yet. «
Comment #5: A thorough understanding of EMT-TF regulation is a prerequisite in defining partial EMT.” Why specifically partial EMT? This has to be elaborated on.
- We agree, both complete and partial EMT should be mentioned, we corrected this in the manuscript.
Reviewer 4 Report
This is an interesting original article entitled “Long non-coding RNAs as potential regulators of EMT-related transcription factors in colorectal cancer – a systematic review and bioinformatics analysis”.
Although, this is an interesting and original article and the results presented are consistent with the text and conclusions, there are some points that need to be addressed before publication.
Comment #1:
The goal of the authors was to investigate the role of lncRNAs (long non-coding RNAs) involved in regulation of the core EMT-TFs (i.e., SNAIL, SLUG, TWIST1, TWIST2, ZEB1 and ZEB2). Thus, the introduction section should contain a brief explanation of these transcription factors, and their main functions.
Comment #2:
The authors mentioned that: “(…) additional relevant studies were manually identified from the references of studies included in this review and review article not included, based on titles concerning the same topic (n = 11)”; however, this sentence does not clarify which criteria were used to include these articles. Why did they have to be manually included? And why didn’t these articles appear in the initial screening?
This concern was not addressed by reading the methods section.
Comment #3:
The discussion section of the paper could be improved. For example, in line 108 authors claim that “ZEB2 was not a sole target of any identified lncRNA”. This is an interesting finding, considering that ZEB1 was identified as a sole target for multiple lncRNAs. ZEB1 and ZEB2 are highly related, but they may have opposing effects and expression patterns tumor biology. This finding should be discussed in the text.
In addition, among the “258 significant correlations of lncRNAs-TF pairs in tumor samples” only ZEB1, ZEB2 and SLUG had strong or very strong correlations. Why do the authors think this is happening? This finding also should be discussed in the text.
Finally, among the only 2 very strong lncRNAs-TF correlations reported, MAGI2-AS3 had a proper discussion section. However, ADAMTS9-AS2 is only mentioned in table 4. Why is that the case?
Author Response
Comment #1: The goal of the authors was to investigate the role of lncRNAs (long non-coding RNAs) involved in regulation of the core EMT-TFs (i.e., SNAIL, SLUG, TWIST1, TWIST2, ZEB1 and ZEB2). Thus, the introduction section should contain a brief explanation of these transcription factors, and their main functions.
- As you suggested, we added a brief description of the main functions of EMT-TFs in both normal tissue (the maintenance of the structure of normal epithelium) and their relevance in disease (activation of EMT, regulation of stemness, therapy resistance, immune evasion, DNA integrity and metabolic reprogramming).
Comment #2: The authors mentioned that: “(…) additional relevant studies were manually identified from the references of studies included in this review and review article not included, based on titles concerning the same topic (n = 11)”; however, this sentence does not clarify which criteria were used to include these articles. Why did they have to be manually included? And why didn’t these articles appear in the initial screening? This concern was not addressed by reading the methods section.
- Thank you for this comment. We added a more detailed description of the search process, providing a clarification regarding this point at the Methods and Results section. Briefly, these articles were not found with the original search because they did not contain any of the following terms: »colorectal«, »colon«, »long non-coding«, »lncRNA«, »SNAIL«, »SNAI1«, »SLUG«, »SNAI2«, »TWIST«, »TWIST1«, »TWIST2«, »ZEB«, »ZEB1«, »ZEB2« in either the title or abstract. They were identified from the references of other articles on the basis of the title indicating the article might be relevant to our study. They were then screened with the rest of the records to assess their relevance based on the inclusion and exclusion criteria.
Comment #3: The discussion section of the paper could be improved. For example, in line 108 authors claim that “ZEB2 was not a sole target of any identified lncRNA”. This is an interesting finding, considering that ZEB1 was identified as a sole target for multiple lncRNAs. ZEB1 and ZEB2 are highly related, but they may have opposing effects and expression patterns tumor biology. This finding should be discussed in the text.
- The fact that none of the lncRNAs were shown to influence only ZEB2 is an interesting finding, indeed. We can only speculatively suggest possible explanations. First, due to the similarity between ZEB1 and ZEB2, it is possible that both are regulated together. Second, it is possible that co-regulation of ZEB2 is a by-stander effect of the regulation of ZEB1. Third, there might co-regulation of TFs similar to that observed in stem cell markers or members of one miRNA family, which can be co-regulated to provide a certain phenotype; it is possible that different EMT-TFs (g., SNAI2 and ZEB2) are co-regulated to activate EMT program. This possibility was added in the text.
Comment #4: In addition, among the “258 significant correlations of lncRNAs-TF pairs in tumor samples” only ZEB1, ZEB2 and SLUG had strong or very strong correlations. Why do the authors - think this is happening? This finding also should be discussed in the text.
- EMT-TFs may have slightly different functions in different types of tissue or may be regulated by other factors in varying intensity. Therefore, it is possible that ZEB1, ZEB2 and SLUG are involved in the regulation of EMT in CRC more strongly than other EMT-TFs, or the correlated lncRNAs have tissue specific-expression influencing these three TFs more strongly than others. This finding was discussed within the text.
Comment #5: Among the only 2 very strong lncRNAs-TF correlations reported, MAGI2-AS3 had a proper discussion section. However, ADAMTS9-AS2 is only mentioned in table 4. Why is that the case?
- We have not discussed ADAMTS9-AS2 because no study has yet evaluated the effects of ADAMTS9-AS2 on either in vivo or in vitro models of CRC. The very strong correlation found through bioinformatics analysis, while interesting, does not necessarily imply a direct regulatory relationship and its role in CRC tissue. Further studies are needed to clarify the role of ADAMTS9-AS2 in CRC.
Round 2
Reviewer 3 Report
The authors have addressed all points raised by this reviewer.
However, some minor points still should be addressed.
Lines 64 / 65 : "Instead, they influence various cellular processes by regulation of their target genes" is a too simplistic description of lncRNA function and has to be rephrased, since the same definition applies to a transcription factor, yet lncRNAs are not transcription factors, but can regulate numerous aspects of gene expression, ranging from regulation of transcription to miRNA sponging etc.
The same applies to "multifunctionality", which is a too generic and actually misleading term to describe that lncRNAs exert their respective function(s) through various molecular mechanisms. A brief recap of these diverse mechanisms would be helpful.
Author Response
Comment #1: Lines 64 / 65 : "Instead, they influence various cellular processes by regulation of their target genes" is a too simplistic description of lncRNA function and has to be rephrased, since the same definition applies to a transcription factor, yet lncRNAs are not transcription factors, but can regulate numerous aspects of gene expression, ranging from regulation of transcription to miRNA sponging etc.
The sentence was rephrased and modes of action of lncRNA added.
Comment #2: The same applies to "multifunctionality", which is a too generic and actually misleading term to describe that lncRNAs exert their respective function(s) through various molecular mechanisms. A brief recap of these diverse mechanisms would be helpful.
The mechanisms of action of lncRNAs were included.